# Novel BCR-Targeting Fusion Proteins for Antigen-Specific Depletion of Alloreactive B Cells in Antibody-Mediated Rejection

**DOI:** 10.3390/cells14181410

**Published:** 2025-09-09

**Authors:** Jing Zhang, Leiyan Wei, Lei Song, Xiaofang Lu, Liang Tan, Xin Li, Li Fu, Qizhi Luo, Xubiao Xie, Yizhou Zou

**Affiliations:** 1Department of Immunology, Xiangya School of Basic Medicine, Central South University, Changsha 410000, China; 2Department of Kidney Transplantation, Center of Organ Transplantation, The Second Xiangya Hospital, Central South University, Changsha 410011, China

**Keywords:** HLA, BCR, NK cells, CDC, ADCC, AMR

## Abstract

Donor-specific anti-HLA antibodies (DSAs) bind to donor vascular endothelial cells and mediate allograft rejection (AMR), but a clinical challenge for which targeted therapeutic options remain limited. We used a multiplexed single-antigen bead (SAB) assay to detect anti-human leukocyte antigen (HLA) antibodies. Based on the antigens which patient’s antibodies aganist to, we developed bivalent HLA-Fc fusion proteins composed of HLA-derived antigenic domains and human IgG1-Fc effector regions (rA24-Fc and rB13-Fc). Specific binding and functional activity of the HLA-Fc proteins were further validated by flow cytometry, ELISA, complement-dependent cytotoxicity (CDC) and antibody-dependent cellular cytotoxicity (ADCC) assays. Our findings demonstrate that the fusion proteins rA24-Fc and rB13-Fc significantly reduced HLA-specific antibody reactivity in vitro. Notably, rA24-Fc and rB13-Fc selectively bound to B-cell hybridomas (e.g., mouse W6/32 cells) expressing membrane immunoglobulins (BCR) which bound to the most HLA class I antigens. Importantly, rA24-Fc and rB13-Fc elicited antigen-specific, Fc-dependent elimination of the specific B-cell hybridomas. This study highlights HLA-Fc fusion proteins as a promising therapeutic strategy for the antigen-specific suppression of depletion of alloreactive B cells through dual cytotoxic mechanisms. This precision targeted to BCR of B cells approach is used to apply to the treatment of antibody-mediated rejection.

## 1. Introduction

Organ transplantation remains the optimal treatment for end-stage organ failure, yet antibody-mediated rejection (AMR) significantly compromises graft survival [1]. This challenge stems largely from donor-specific anti-HLA antibodies (DSAs), a major cause of graft loss affecting 3–30% of recipients [2,3,4,5,6]. DSAs arise due to significant genetic polymorphism in HLA loci, leading to frequent donor–recipient mismatches that trigger anti-donor HLA antibody production [1]. HLA class I antigens (including HLA-A, HLA-B, and HLA-C) are expressed on the surface of all nucleated cells, while HLA class II antigens are primarily restricted to antigen-presenting cells and thymic epithelial cells [2]. Structurally, HLA-I molecules comprise a polymorphic α-chain (encoded by exson 2-4 on human chromosome 6) and β_2_-microglobulin (encoded on chromosome 15). The extracellular portion of the α-chain consists of three domains, α1 and α2, which form the peptide-binding groove, and the α3 domain, which binds CD8 and facilitates T-cell recognition [3]. The structure of fragment crystallizable (Fc) region of immunoglobulin G (IgG), located at the C-terminus of the heavy chain, includes the CH_2_ and CH_3_ domains [4]. This structural arrangement enables the Fc region to interact with various receptors—such as FcγR, FcαR, and FcRn—thereby playing essential roles in immune regulation, antibody-dependent cellular cytotoxicity (ADCC), complement-dependent cytotoxicity (CDC), and the extension of antibody half-life [7].

Alloreactive B cells (allo-B cells) expressing donor-HLA-specific B cell receptors (BCRs) are pivotal in DSA generation in a given patient [8]. Interestingly, the extracellular domain of the allo-B cell BCR exhibits specificity identical to the secreted DSA, serving as both a molecular signature identifying pathogenic clones and a therapeutic target for selective clearance [8,9].

The Allele Frequency Net Database (AFND) was used to summarize the common HLA-A/B loci alleles and their frequencies. The carriers of alleles such as HLA-A*24:02 and HLA-B*13:02 had a relatively high frequency (A24:02, 34%; B13:02, 21%; Allele Frequency Net) in the population all over the world. To enable targeted therapy, we designed recombinant HLA-Fc fusion proteins targeting the specific B cells with the BCR for binding. Therefore, we developed BCR-targeting fusion proteins HLA-A*24:02 (rA24-Fc) and HLA-B*13:02 (rB13-Fc). To specifically eliminate Allo-B lymphocytes that mainly produce anti-A24 or anti-B13 antibodies, we developed an rA24-Fc or rB13-Fc fusion protein, whose N-terminal is the extracellular segment of the HLA-A24 or HLA-B13 antigen, which can bind to the BCR on the surface of Allo-B cells that produce anti-A24 and an-B13 antibodies, and whose C-terminal is the Fc segment of human IgG1, which can bind to the Fc receptors on the surface of immune cells, thereby activating immune response. Natural Killer (NK) cells can specifically eliminate Allo-B cells through mechanisms like ADCC and CDC [10,11]. The soluble rA24-Fc and rB13-Fc fusion proteins can specifically eliminate Allo-B cells that produce anti-A24 and anti-B13 antibodies, without causing damage to other B cells [12,13]. Moreover, it is possible to design additional HLA-FC fusion proteins for patients who have produced other HLA antibodies, providing a novel approach for specifically eliminating Allo-B cells that produce HLA antibodies after kidney transplantation.

## 2. Materials and Methods

### 2.1. Clinical Samples

Serum samples were collected from four renal transplant recipients exhibiting impaired graft function, defined by a serum creatinine level exceeding 500 μmol/L, and one healthy control individual with no history of transfusion, pregnancy, or transplantation to rule out pre-existing anti-HLA antibodies. All samples were obtained at the Second Xiangya Hospital of Central South University between December 2019 and May 2022, prior to the initiation of any desensitization therapy. Following coagulation, serum was isolated, aliquoted, and stored at −20 °C. This study received approval from the Institutional Review Board of the Second Xiangya Hospital of Central South University, with the approval code 2020 Lunjian Clinical Trial No. K005 and approval date 30 October 2020.

### 2.2. Cell Culture

Hybridoma W6/32 cells (producing anti-HLA-I mAb, ATCC), 9B10 cells (producing anti-MICA mAb, ATCC), and NKL cells (provided by M. J. Robertson, Indiana University School of Medicine, Indianapolis, Indiana) were cultured in RPMI 1640 medium (Gibco, Cat# 11875500BT, Grand Isle, New York, NY, USA) supplemented with 10% FBS (Gibco, Cat# 10099141, Calif, USA). NKL medium was further supplemented with 10 ng/mL IL-2 (PeproTech, Cat# 200-02-50UG, Cranbury, NJ, USA). Suspension HEK293 cells were cultured in serum-free SMM 293-TII medium (Sino Biological, Cat# M293TII, Beijing, China). All cell lines were maintained at 37 °C in a humidified 5% CO_2_ atmosphere. In the complement-dependent cytotoxicity experiment, cells need to be incubated with fresh rabbit serum for W632 cells and 9B10 cells.

### 2.3. Fusion Protein Construction

Recombinant pcDNA3.4 plasmids encoding HLA-A*24:02-ECD-Fc or HLA-B*13:02-ECD-Fc were constructed as follows: Core nucleotide sequences were synthesized by Tsingke Biotechnology (Beijing, China). HLA-A*24:02 extracellular domain (ECD) amplified with primers 1F/1R. HLA-B*13:02 ECD amplified with primers 3F/3R. IgG1-Fc domain amplified from hIgG1-H-pCMV5.1 template using primers: A24-Fc: 2F/2R, B13-Fc: 4F/4R, Overlap Extension PCR: A24-ECD + Fc fragments → A24-targeting fragment (primers 1F/2R), B13-ECD + Fc fragments → B13-targeting fragment (primers 3F/4R). Fragments were ligated into pcDNA3.4 via XbaI/AgeI restriction sites (NEB: R0145V, R0552S, Ipswich, MA, USA) using T4 DNA ligase (Thermo, #01001207, Waltham, MA, USA). Ligation products were transformed into E. coli DH5α competent cells (TransGen, #CD201-01, Beijing, China). Plasmid DNA from positive clones was sequenced for verification prior to use.
**Primer****Sequences**1FGGACTCTAGAGCTATGGGCCTG1RACTTCCTCCTCCTCCACTCCCACCCCCACCGGAGCCGCCACCACCCCATCTCAGTGTCAGGGG2FGGTGGTGGCGGCTCCGGTGGGGGTGGGAGTGGAGGAGGAGGAAGTACTCACACATGCCCACCG2RACTAACCGGTTTATTTCCCGGGAGACAG3FGGACTCTAGAGCCATGGGCCTG3RACTTCCTCCTCCTCCACTCCCACCCCCACCGGAGCCGCCACCACCCCATCTCAGGGT4FGGTGGTGGCGGCTCCGGTGGGGGTGGGAGTGGAGGAGGAGGAAGTACTCACACATGCCCAC4RACTAACCGGTCTATTTCCCGGGAGACAG


### 2.4. Protein Expression and Purification

Recombinant HLA-A*24:02-ECD-Fc and HLA-B*13:02-ECD-Fc proteins were expressed in suspension HEK293 cells via PEI-mediated transfection (Polysciences, #23966, Warrington, PA, USA) using a 1:3 (*w*/*w*) plasmid: PEI ratio incubated for 15 min at RT, with final DNA concentration at 1 μg/mL. Culture supernatants harvested 72–96 h post-transfection were diluted 1:3 in PBS and subjected to Protein A affinity chromatography (Cytiva, Marlborough, MA, USA). Bound proteins were eluted with low-pH buffer (pH 2.2–2.8), immediately neutralized with 1M Tris-HCl (pH 8.8; Beyotime, #P0012A-2, Shanghai, China), and dialyzed against pH 7.4 buffer until clarification. His-tagged constructs were further purified by Ni-NTA chromatography (Bestchrom, #AA0052, Pinghu, China) with 0.5M imidazole elution (Sigma, #I5513, Kawasaki, Japan). All proteins were concentrated using 30 kDa MWCO Amicon^®^ Ultra-4 centrifugal filters (Merck, #UFC803024, Darmstadt, Germany) and stored at −80 °C.

### 2.5. Coomassie Brilliant Blue Staining

Purified recombinant proteins (rA24-Fc, rB13-Fc) were analyzed by SDS-PAGE. Samples (30 μL) were mixed with 10 μL 4 × Laemmli buffer (Solarbio, #P1015, Beijing, China), denatured at 95 °C for 5 min, and separated on 12% Tris-glycine gels with 5% stacking layers. Following electrophoresis, gels were stained with Coomassie Brilliant Blue R-250 (Sigma, #B0149, Kawasaki, Japan) for 30 min at RT with gentle agitation, then destained overnight in methanol:acetic acid solution (40:10 *v*/*v*).

### 2.6. Western Blot Analysis

Following electrophoresis, proteins were transferred onto PVDF membranes (Merck Millipore, #0000187588). Membranes were blocked with 5% skim milk in PBS for 2 h at 37 °C. After washing with 0.1% PBST (Biosharp, #21063642, Hefei, China), blots were incubated with either: HRP-conjugated anti-His antibody (Abmart, #M20020, Shanghai, China), HRP-conjugated anti-human IgG-Fc antibody (Jackson ImmunoResearch, #109-035-098, Lancaster, PA, USA) for 1 h at 37 °C. Following three PBST washes, protein bands were detected using ECL substrate (APPLYGEN, #P1050, San Francisco, CA, USA) on a Tanon 5200Multi chemiluminescence imaging system.

### 2.7. ELISA for Functional Validation of Recombinant Proteins

Polystyrene plates (BIOFIL, #FEP-101-896, Guangzhou, China) were coated with 1 μg/mL recombinant proteins (A24, B13, or BSA control; GENVIEW, #9048-46-8, Houston, TX, USA) or W6/32 antibody in PBS at 4 °C overnight. After blocking with 5% skim milk/PBS (4 °C overnight), plates were washed thrice with 0.05% PBST and incubated with A24/B13 plates: 100 μL/well of diluted primary antibodies, 1:10 serum in PBS, or cell culture supernatants (37 °C, 2 h); W6/32 plates: serial dilutions of rA24-Fc, rB13-Fc, or hIgG1-H control (starting at 1 μg/mL, 37 °C, 2 h). All wells were subsequently probed with HRP-conjugated antibodies (anti-His mAbs [Jackson, #115-035-071, Lancaster, PA, USA] or anti-human IgG for sera/proteins; 1:4000 dilution, 37 °C, 2 h). Following PBST washes, reactions were developed with TMB substrate (Solarbio, #PR1200; 100 μL/well, 37 °C, 15 min dark), terminated with 50 μL 2M H_2_SO_4_, and read at 450 nm (AMR-100 microplate reader).

### 2.8. Flow Cytometric Analysis of BCR Binding

W6/32 hybridoma cells (1 × 10^6^/tube) were resuspended in PBS/4% FBS. Cells were incubated with 1 μg/mL of rA24-Fc, rB13-Fc, hIgG1-H (control) at 4 °C for 30 min. After two washes with PBS/4% FBS, PE-conjugated goat anti-human IgG (1:100; BioLegend, #409304, San Diego, CA, USA) was added at 4 °C for 30 min. Washed cells were analyzed by flow cytometry (CYTEK^TM^ DxP Athena^TM^, USA).

### 2.9. Complement-Dependent Cytotoxicity (CDC) Assay

W6/32 or 9B10 hybridoma cells were incubated with serially diluted BCR-targeting fusion protein in fresh rabbit complement serum (Cedarlane, #CL4051, Burlington, ON, Canada) at 37 °C for 2 h. Cells were washed twice with PBS/4% FBS, then stained with 5 μg/mL acridine orange (AO; Macklin, #C13300058, Shanghai, China) and propidium iodide (PI; Biofroxx, #1246MG01D, Einhausen, Germany) in dark at room temperature for 10 min. After final washes, viability was quantified by fluorescence microscopy (Leica DMi8, Leica Microsystems CMS GmbH, Mannheim, Germany).

### 2.10. Antibody-Dependent Cellular Cytotoxicity (ADCC) Assay

CFSE-stained target cells (W6/32 or 9B10 hybridomas; 5 × 10^4^/well) were co-cultured with NK cells (effector: target = 8:1) in 96-well plates with BCR-targeting fusion proteins (0–20 μg/mL). After 4 h at 37 °C, cells were stained with 7-AAD (BD Pharmingen, #559925, San Diego, CA, USA) and analyzed by flow cytometry (CYTEK^TM^ DxP Athena^TM^, USA). Cytotoxicity was calculated as the percent of CFSE^+^7-AAD^+^ cells. CFSE protocol: Cells stained with 5μM CFSE (BD, #565082, Franklin Lakes, NJ, USA) in PBS/4% FBS at 37 °C for 30 min, washed thrice before use.

### 2.11. Statistical Analysis

Data from ≥3 independent experiments are presented as mean ± standard deviation. Normality was confirmed by the Kolmogorov–Smirnov test. Comparisons were made using Student’s *t*-test, one-way analysis of variance (LSD post hoc test), and chi-square test. The significance threshold was *p* < 0.05 (GraphPad Prism 9.5).

## 3. Results

### 3.1. Design and Production of HLA-Fc Proteins

Anti-HLA antibodies were detected in the sera of four renal transplant recipients and one healthy control using LABScreen™ Single Antigen Beads (LABScreen Single Antigen ExPlex, Thermo Fisher Scientific, Waltham, MA, USA). The analysis revealed that serum#01 harbored high-titer anti-HLA-A*24:02 antibodies (MFI > 12,000), and serum#02 exhibited anti-HLA-B*13:02 reactivity (MFI > 12,000), while no HLA-A/B antibodies were detected in the healthy control sample (Figure 1A). Serum containing anti-A24/B13 antibodies also demonstrated binding to epitopes on other cross-reactive beads, indicating that different anti-HLA-I antibodies can share public epitopes. This cross-reactivity provides an opportunity to evaluate how epitopes present on HLA-A24:02 and B13:02 may extend to other HLA-I antigens.

According to our previous research, HLA-I can be coupled with the Fc region of IgG to form a bilaterally symmetrical antibody-like homodimer (Figure 1B). Structural modeling using SWISS-MODEL predicted a well-folded structure with distinct domains and no spatial hindrance between the HLA-ECD and Fc modules (Figure 1C). As displayed in the 3D model, the HLA-ECD is positioned to engage the B cell receptor (BCR) on target allo-B cells, while the Fc domain facilitates protein purification, detection, and mediates cytotoxic effector functions. We expect that this fusion protein will bind to surface anti-HLA immunoglobulin of B cells and trigger cytotoxicity through Fc-dependent effector NKL functions. Building upon our prior verification that patients with high-titer anti-HLA-A*24:02 and anti-HLA-B*13:02 serum antibodies. A human IgG1-H-His protein was used as a control (Figure 1D). rA24Fc and rB13Fc fusion proteins comprise N-terminal HLA extracellular domain (HLA-ECD) module and a C-terminal human IgG1 Fc domain. Expression vectors for rA24-Fc and rB13-Fc were constructed with the following architecture (N- to C-terminus): Kozak sequence, signal peptide, the first flexible linker, β_2_-microglobulin, a second flexible linker, HLA class I heavy chain extracellular domains (α1-α3), and the human IgG1 Fc domain (Figure 1D). Recombinant plasmids were transfected into suspension HEK293 cells. Proteins secreted into the supernatant were purified via Protein A affinity chromatography, yielding approximately 10 mg per batch. Coomassie Brilliant Blue staining (Figure 1E) and Western blot analysis (Figure 1F) confirmed that the purified proteins migrated at the expected molecular weight (90 kDa) with high purity, along with the hIgG1-H control (55 kDa). The expected design was consistent, indicating that the recombinant proteins A24-Fc and B13-Fc were successfully prepared.

### 3.2. The rA24-Fc and rB13-Fc Exhibits Binding Affinity for Both W6/32 Hybridoma Cells and Their Secreted Antibodies

To verify the functions of rA24-Fc and rB13-Fc proteins, we used a broad-spectrum HLA-I class monoclonal antibody producing hybrodoma (W6/32, ATCC).It is one of the most common monoclonal antibodies used for identifying human HLA-I class molecules. It can recognize the conformational epitopes on the complete HLA molecule that contain β2m and the heavy chain [14,15]. ELISA plates coated with W6/32 mAb were incubated with recombinant proteins (1 μg/mL; rA24-Fc, rB13-Fc, Fc), followed by HRP-conjugated goat anti-human IgG. Results demonstrated significantly higher binding affinity of both rA24-Fc and rB13-Fc for the immobilized W6/32 mAb compared to hIgG1-H (Figure 2A). W6/32 exhibited dose-dependent recognition of the targeting bodies, while showing no reactivity to Fc across tested concentrations (Figure 2A). Based on prior screening, serum #01 and serum #02 contained high-titer antibodies specific to HLA-A*24:02 and HLA-B*13:02, respectively. Normal healthy serum (serum # normal), lacking significant HLA class I reactivity, served as a control. All sera were diluted 1:10 for use as primary antibodies in indirect ELISA. Results confirmed that serum #01 (anti-HLA-A*24:02) showed robust reactivity against the rA24-Fc protein compared to normal sample (Figure 2B). Similarly, serum #02 (anti-HLA-B*13:02) displayed strong binding to the rB13-Fc protein (Figure 2B). These findings indicate that the purified HLA recombinant proteins retain high functional activity.

Leveraging the shared epitope specificity between the BCR and secreted antibody of a B cells, we incubated W6/32 hybridoma cells and 9B10 hybridoma cells (secretory resistance mouse hybridoma cells with MICA monoclonal antibody as mock group) with rA24-Fc, rB13-Fc, or hIgG1-H and analyzed binding by flow cytometry (Figure 2C). Both rA24-Fc and rB13-Fc showed significant, specific binding to the surface of W6/32 hybridoma cells compared to the hIgG1-H control and mock (9B10 hybridoma cells) (Figure 2C).

### 3.3. BCR-Targeting Fusion Protein rA24-Fc and rB13-Fc Mediates Specific Hybridoma Cell Lysis via Complement Activation

To test the binding capability of rA24-Fc and rB13-Fc to W6/32 hybridoma cells, we hypothesized that these fusion proteins would induce specific target cell lysis with minimal off-target effects. We assessed the killing efficacy of the BCR-targeting fusion proteins against anti-HLA class I antibody-producing hybridoma cells using complement-dependent cytotoxicity (CDC) assays. W6/32 hybridoma cells were incubated with the fusion proteins along with diluted rabbit serum (as a source of complement) for 1 h at 37 °C in the dark. Cell viability was determined by acridine orange (AO)/propidium iodide (PI) dual fluorescence staining: viable cells fluoresce green (AO^+^), while dead cells fluoresce red (PI^+^). The percentage of killed cells was calculated by counting 200 cells under fluorescence microscopy.

As expected, untreated control W6/32 cells showed high viability (Figure 3A,B). Treatment with 5 μg/mL of either targeting fusion protein induced significant lysis; rA24-Fc treatment resulted in 55% ± 6.51% killing (*p* < 0.001), and rB13-Fc treatment resulted in 86% ± 0.76% killing (*p* < 0.001). The ‘Mock’ control, represented by 9B10 cells (which secrete anti-MICA antibodies but not anti-HLA antibodies), showed no obvious killing mediated by rA24-Fc or rB13-Fc. These results demonstrate that the BCR-targeting fusion proteins achieve potent and specific lysis of targeting BCR specific hybridoma cells (W6/32) in presence of complement activation.

### 3.4. BCR-Targeting Fusion Proteins Mediates NK Cell-Dependent Lysis of W6/32 Hybridoma Cells

Building on our demonstration that BCR-targeting fusion proteins specifically eliminate W6/32 cells via complement activation, we hypothesized these fusion proteins could also serve as molecular bridges to direct NK cell-mediated cytotoxicity against target cells. To test this, we assessed antibody-dependent cellular cytotoxicity (ADCC) using W6/32 (cognate target) and 9B10 (non-target control) hybridoma cells.

Target cells were labeled with CFSE and co-cultured with purified NK cells at an 8:1 effector-to-target (E:T) ratio. Cell death was quantified by flow cytometry detection of apoptotic cells (CFSE^+^7-AAD^+^). At concentrations of 1, 10, and 20 μg/mL, A24-targeting fusion protein mediated 28%, 50%, and 72% NK cell lysis of W6/32 cells (Figure 4A), also showing dose-dependent efficacy (significantly higher than control [22%]). Under identical conditions, rB13-Fc mediated NK cell lysis of W6/32 cells at 19.69%, 24.83%, and 48.07%, respectively (Figure 4B), demonstrating significant dose-dependent killing (*p* < 0.0001 vs. control [8.2%]). Neither BCR-targeting fusion proteins induced significant NK-mediated lysis of 9B10 cells (which lack anti-HLA class I antibody production) at any concentration tested. These results demonstrate that BCR-targeting fusion proteins specifically redirect NK cell cytotoxicity against cognate target hybridoma cells (W6/32 cells) while sparing mock (9B10 cells).

## 4. Discussion

Although significant advances have been made in immunosuppressive therapy in recent years, renal allograft rejection remains a major challenge limiting long-term graft survival. Antibody-mediated rejection (AMR) is a principal cause of chronic graft loss, and current strategies—which largely target T-cell responses—exhibit limited efficacy against AMR [14]. Furthermore, long-term immunosuppression increases the risk of infections and malignancy due to broad immune suppression [15]. Current AMR management includes plasmapheresis, immunoadsorption, intravenous immunoglobulin (IVIg), anti-CD20 humanized monoclonal antibody (e.g., rituximab), complement inhibitors, and proteasome inhibitors [16]. However, these approaches are generally non-specific, aiming to remove circulating antibodies without any selectivity, causing infection. For instance, combining high-dose IVIg with anti-CD20 therapy has demonstrated superior outcomes compared to IVIg monotherapy in living or deceased donor kidney transplant recipients [17,18,19,20,21]. Emerging fusion proteins such as proteasome inhibitors (e.g., bortezomib), complement blockers (e.g., eculizumab), anti-IL-6 receptor antibodies, and bacterial enzymes like imlifidase are also under investigation. Bortezomib targets antibody-producing plasma cells by inhibiting proteasome activity and has been used alongside standard therapy for acute AMR [20]. Eculizumab, a monoclonal antibody against complement C5, prevents membrane attack complex formation and has been employed pre-transplant in sensitized patients to inhibit early complement-mediated injury [17]. The effects of these drugs are non-specific, and they can harm many types of immune cells, thereby increasing the risk of other concurrent diseases in the body. For example, bortezomib may cause side effects such as gastrointestinal toxicity, thrombocytopenia, and peripheral neuropathy. Additionally, eculizumab as a complement inhibitor may increase the risk of infection. Because rituximab affects main B cells, it may reduce the body’s resistance to infection and increase the risk of bacterial, viral or fungal infections.

Recently, cellular therapies have opened new avenues for AMR treatment. Chimeric antigen receptor T-cell (CAR-T) technology, initially developed for autoimmune disorders, has been adapted for transplant-related alloimmunity. For example, T cells engineered with a chimeric HLA antibody receptor (A2-CHAR) against HLA-A2 effectively eliminate HLA-A2-specific antibody-producing B cells via granzyme B-mediated cytotoxicity. In immunodeficient mouse models, A2-CHAR-T cells cleared target B cells even within the bone marrow in a dose-dependent manner. Similarly, regulatory T cells expressing an HLA-A2-specific CAR (A2-CAR-Tregs) have shown enhanced efficacy in preventing graft-versus-host disease, underscoring the potential of CAR-based strategies in promoting tolerance [22].

Emerging strategies such as CAR-T and CAR-Treg cells have shown promise in targeting alloreactive B cells (Allo-B) and promoting tolerance. We developed HLA-Fc fusion proteins (rA24-Fc and rB13-Fc) consisting of the extracellular domain of HLA-I molecules fused to an IgG1 Fc segment (Figure 5). These molecules selectively engage alloreactive B cells via BCR recognition and elicit cytotoxic effects through CDC and ADCC, as demonstrated by efficient elimination of W6/32 hybridoma cells secreting anti-HLA-I antibodies in vitro. The modular design allows for substitutions of the HLA moiety, enabling the broad targeting of alloreactive B cells with distinct HLA specificities, thereby supporting a precision medicine approach with potentially reduced off-target effects. Our strategy is designed to eliminate only the small fraction of B cells with specificity for DSA. The vast majority of B cells responsible for protective immunity against pathogens remain intact. This preserved immune competence could translate into a dramatically improved safety profile, allowing for safer long-term or repeated administration to control alloimmune responses.

Key advantages of this approach include high specificity, dual cytotoxic mechanisms (CDC and ADCC), and adaptable HLA domains for personalized application. What should be done if the DSA produced by the patient is not anti-A24 or anti-B13? Specific fusion proteins of HLA antigen Fc can be designed based on the specificity of the antibodies that target the receptors, but the desensitization treatment for HLA-II class antibodies cannot be expanded. However, several limitations must be acknowledged, including the use of hybridoma models rather than patient-derived B cells, lack of in vivo validation, potential neutralization by pre-existing antibodies, and inability to target plasma cells. First, validation was performed primarily in vitro using a hybridoma cell line; future work should employ primary human B cells or immortalized alloreactive B cell clones. Second, in vivo efficacy remains to be established. Humanized mouse models reconstituted with patient-derived immune components would better recapitulate the clinical AMR milieu. Third, pre-existing anti-HLA antibodies in patients may neutralize administered fusion proteins. Combination strategies—such as pre-treatment with plasmapheresis, immunoadsorption, or IgG-degrading enzymes—may enhance bioavailability and efficacy.

Despite these challenges, HLA-Fc fusion proteins represent a promising targeted therapeutic strategy for the specific depletion of alloreactive B cells, potentially offering long-term benefit for transplant recipients. This work provides a foundational rationale for further development of antigen-specific immunotherapies against AMR.

Hybridoma-based in vitro models potentially lacking patient-derived B-cell complexity, requiring future validation with sensitized recipient-derived B cells and insufficient exploration of circulating antibody neutralization effects. AMR causes significant graft failure despite 120,000 annual transplants [16], with only 100,000 functional grafts globally [17]. Lifelong immunosuppressants—primarily T-cell targeted—fail to control B-cell-mediated AMR [22,23] while increasing infection/cancer risks and graft toxicity [18]. No FDA-approved AMR therapies exist; current non-specific approaches (plasma exchange, IVIG, rituximab, proteasome inhibitors) show limited efficacy [19,21,24,25,26,27,28], with randomized trials (RITUX ERAH [27], BORTEJECT [28]) demonstrating no clinical benefit. Targeting alloreactive B cells (Allo-B)—precursors of donor-specific antibody (DSA)-producing plasma cells—leverages our hypothesis that DSA presence necessitates antigen-specific B-cell clones, with the BCR serving as their unique surface marker. While CAR-Treg and CAR-T [29,30,31] technologies show promise, and oncology utilizes antigen-IgFc fusion proteins (“peptide antibodies”) for precision targeting, we selected Allo-B elimination to establish humoral tolerance. Screening sera from renal transplant recipients identified immunodominant HLA-A*24:02/B*13:02 epitopes. However, we explicitly acknowledge that our approach, as a BCR-targeting fusion proteins, does not directly target long-lived plasma cells, as they are terminally differentiated and have downregulated surface BCR expression. Consequently, pre-existing DSA produced by these cells will persist in the circulation initially and could indeed obscure early efficacy signals in clinical trials. A rapid decline in DSA might not be observed. We propose that the most logical and potent clinical application of our agent would be in combination with plasma cell-directed therapies. Agents such as proteasome inhibitors (e.g., bortezomib) or anti-CD38 monoclonal antibodies (e.g., daratumumab) are specifically designed to deplete plasma cells. A sequential strategy could be envisioned: first, using a plasma cell-depleting agent to rapidly reduce the load of mature DSA-producing cells, followed by our antigen-specific therapy to prevent the relapse of DSA by eliminating the donor-specific memory and naive B cells that would otherwise replenish the plasma cell. Finally, the produced antibodies are removed through plasma exchange and antibody adsorption. This synergistic approach could lead to deeper and more durable reductions in DSA.

## 5. Conclusions

We designed BCR-targeting fusion proteins by fusing HLA modules to human IgG1 Fc. These rA24-Fc and rB13-Fc bound membrane BCR and secreted antibodies dose-dependently, mediated specific CDC lysis of W6/32 cells, and triggered potent NK cell-directed ADCC, while sparing non-target 9B10 cells. The modular design permits HLA domain substitution to target diverse Allo-B specificities. Despite these challenges, HLA-Fc fusion proteins represent a promising precision medicine strategy for depleting Allo-B cells—precursors of plasma cells that produce donor-specific antibodies. Future work should focus on in vivo evaluation using humanized models, development of bispecific constructs, and combination therapies with IgG-degrading enzymes or desensitization protocols. This study establishes a foundational framework for antigen-specific clearance of pathogenic B cells in transplantation.

## Figures and Tables

**Figure 1 cells-14-01410-f001:**
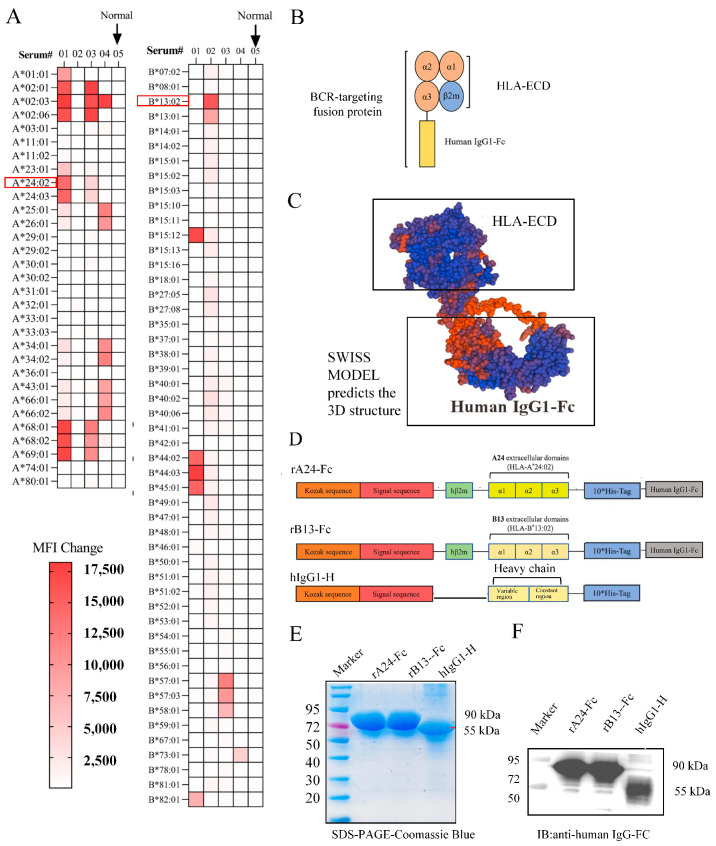
Design, preparation, and production of recombinant HLA-A*24:02 Fc and HLA-B*13:02 Fc fusion protein. (**A**) Each row corresponds to a distinct bead coated with a unique HLA-I single antigen. The intensity of red color indicates the level of antibody binding, with darker red representing higher mean fluorescence intensity (MFI) values. Sera 1–4 were obtained from kidney transplant recipients; serum N was collected from a healthy individual as a negative control. (**B**) Schematic structure of the BCR-targeting fusion protein. The N-terminal module consists of incorporating the HLA-A*24:02 or HLA-B*13:02 extracellular domain (ECD), enabling specific binding to the B cell receptor (BCR) on alloreactive B cells (Allo-B cells). The C-terminal module comprises the human IgG1 Fc domain, mediating immune effector functions. (**C**) 3D Structural modeling using SWISS-MODEL predicted a well-folded structure with distinct domains. (**D**) Gene construct design for the BCR-targeting fusion proteins (rA24-Fc and rB13-Fc) and hIgG1-H. Key elements (5′ to 3′): Xba I restriction site, Kozak sequence, signal peptide, HLA antigen ECD (A24:02 or B13:02), linker sequence, IgG1 Fc domain, Age I restriction site. (**E**) Coomassie Brilliant Blue-stained SDS-PAGE gel. (**F**) Western blot analysis.

**Figure 2 cells-14-01410-f002:**
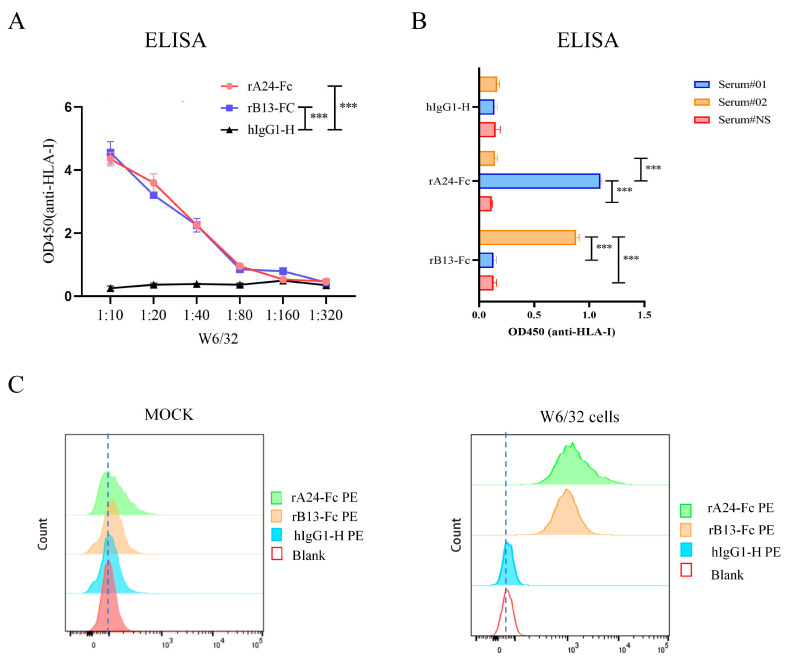
Functional validation of recombinant HLA-A*24:02 and HLA-B*13:02 Fc fusion protein. (**A**) ELISA assessing binding of rA24-Fc, rB13-Fc to W6/32 monoclonal antibody. H-IgG1-H represents the control group). (**B**) The rA24-Fc reacts with the serum #01. The rB13-Fc react with serum #02, and all of these will react when compared to the control serum #normal. Supernatants from cultured cells were used as primary antibody sources. (**C**) Flow cytometric analysis of rA24-Fc and rB13-Fc to surface BCR on W6/32 hybridoma cells, quantitative analysis of binding. Isotype control: Mock represents 9B10 cells. Data analyzed by ANOVA; significance denoted as *** *p* < 0.001. Each experiment was repeated three times (*n* = 3).

**Figure 3 cells-14-01410-f003:**
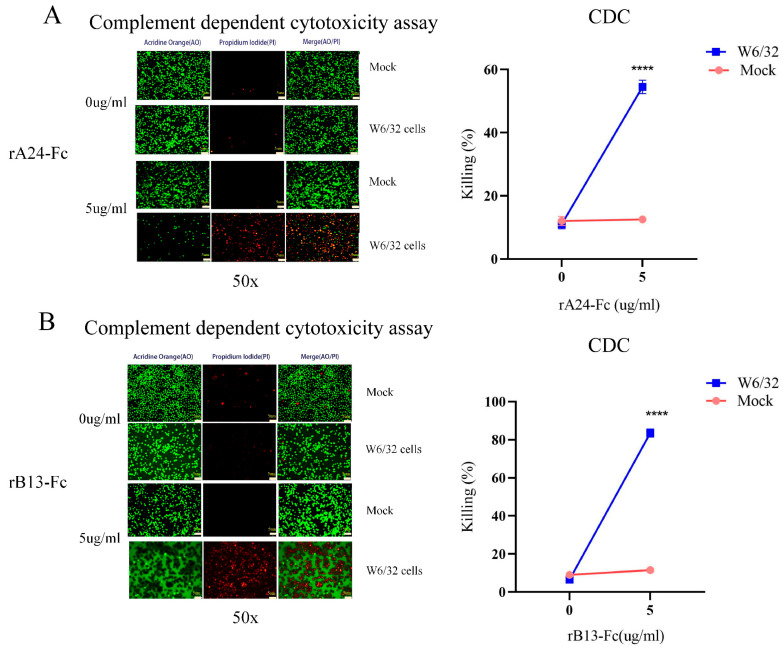
rA24-Fc and rB13-Fc induce specific complement-mediated lysis of W6/32 and “mock” 9B10 hybridoma cells. Cell viability was assessed using acridine orange (AO)/propidium iodide (PI) dual fluorescence staining. Viable cells fluoresce green (AO^+^), while dead cells fluoresce red (PI^+^). (Mock’ refers to the use of 9B10 cells as a non-target negative control.) (**A**) Representative AO/PI staining images of W6/32 cells (target) or 9B10 cells (negative control) incubated with rA24-Fc (0 and 5 µg/mL) in the presence of complement. Quantitative analysis of complement-mediated lysis for rA24-Fc against W6/32 and 9B10 cells (corresponding to panel A). (**B**) Representative AO/PI staining images of W6/32 cells (target) or 9B10 cells (negative control) incubated with rB13-Fc (0 and 5 µg/mL) in the presence of complement. Quantitative analysis of complement-mediated lysis for rB13-Fc against W6/32 and 9B10 cells (corresponding to panel C). Data were analyzed by ANOVA. **** *p* < 0.0001. Each experiment was repeated three times (*n* = 3).

**Figure 4 cells-14-01410-f004:**
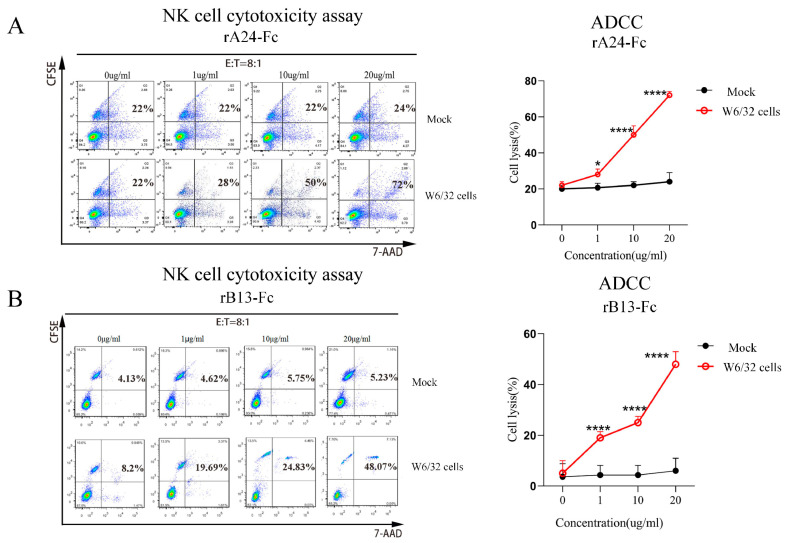
BCR-targeting fusion proteins mediate NK cell-dependent lysis of W6/32 hybridoma cells via ADCC. (**A**) Flow cytometric analysis of NK cell (low square)-mediated killing of CFSE-labeled W6/32 cells (target) (up square) and mock (Mock’ refers to the use of 9B10 cells as a non-target negative control.) co-cultured at an 8:1 effector-to-target (E:T) ratio in the presence of increasing concentrations (0, 1, 10, 20 µg/mL) of rA24-Fc. Quantitative analysis of specific lysis of W6/32 and 9B10 cells mediated by the rA24-Fc. (**B**) Flow cytometric analysis of NK cell-mediated killing of CFSE-labeled W6/32 cells (target) and 9B10 cells (control) at an 8:1 E:T ratio in the presence of increasing concentrations (0, 1, 10, 20 µg/mL) of B3Fc. Quantitative analysis of specific lysis of W6/32 and 9B10 cells mediated by the rB13-Fc. Data were analyzed by an unpaired *t*-test. * *p* < 0.05, **** *p* < 0.0001. Each experiment was repeated three times (*n* = 3).

**Figure 5 cells-14-01410-f005:**
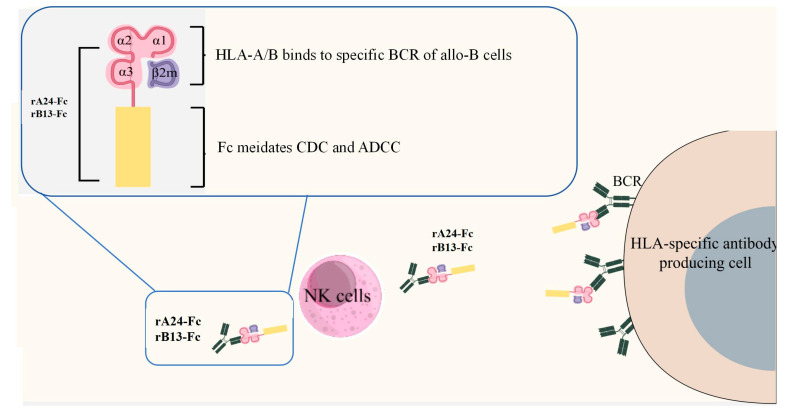
Structure and mechanism of HLA-Fc fusion proteins (rA24-Fc and rB13-Fc). The soluble rA24-Fc and rB13-Fc fusion proteins consist of an HLA-derived antigen-binding domain (specific for HLA-A*24:02 or HLA-B*13:02) fused to the Fc domain of human IgG1. The Fc domain mediates homodimerization of the fusion protein. The HLA ECD binds specifically to the BCR on allo-B cells. Simultaneously, the Fc domain engages Fc receptor on NK cells, triggering ADCC. Additionally, upon binding to the BCR, the Fc domain undergoes a conformational change that enables complement binding and activation, leading to CDC. Together, these mechanisms promote specific elimination of allo-B cells.

## Data Availability

The datasets generated and analyzed during the current study are not publicly available due to confidentiality but are available from the corresponding author on reasonable request.

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
