# Peer review of "Novel BCR-Targeting Fusion Proteins for Antigen-Specific Depletion of Alloreactive B Cells in Antibody-Mediated Rejection"

_cells, 2025, doi:10.3390/cells14181410_

Round 1

Reviewer 1 Report

Comments and Suggestions for Authors

I believe this manuscript is highly relevant contribution to the field of transplantation immunology. The concept of BCR-targeting fusion proteins for antigen-specific depletion of pathogenic B cells in AMR is well articulated and supported by convincing in vitro data. The methodological rigor—including careful biochemical characterization, structural modeling, and functional validation using both CDC and ADCC assays—strengthens the credibility of the findings.

Major comments

While the selective elimination of donor HLA–specific B cells is conceptually appealing and may reduce off-target immunosuppression, it would strengthen the clinical perspective if you could discuss more explicitly how this approach might provide tangible benefits over current B-cell–depleting therapies such as rituximab. Rituximab, which depletes the majority of circulating B cells, has shown only limited efficacy in post-transplant AMR in randomized trials. Clarifying why a narrower, antigen-specific depletion strategy could yield superior outcomes—whether by preserving protective immunity, enabling repeated dosing, or targeting pathogenic clones more efficiently—would help readers appreciate the translational potential and realistic clinical impact of your platform.

Minor comments

The discussion of study limitations could be further enriched to help readers envision the steps toward clinical application. In particular, since plasma cells lack surface BCR, they are not direct targets of this approach. It would be valuable to outline potential complementary strategies, such as combination with plasma cell–directed agents (e.g., proteasome inhibitors or anti-CD38 antibodies). In a real-world setting, evaluating the clinical impact of BCR-targeting proteins might be complicated by the persistence of donor-specific antibodies from long-lived plasma cells. Discussing how such residual antibody production could obscure early efficacy signals would help set realistic expectations.

Author Response

Author's Reply to the Review Report (Reviewer 1)

I believe this manuscript is highly relevant contribution to the field of transplantation immunology. The concept of BCR-targeting fusion proteins for antigen-specific depletion of pathogenic B cells in AMR is well articulated and supported by convincing in vitro data. The methodological rigor—including careful biochemical characterization, structural modeling, and functional validation using both CDC and ADCC assays—strengthens the credibility of the findings.

Response: Thank you for your recognition and encouragement of our article.

Major comments

While the selective elimination of donor HLA–specific B cells is conceptually appealing and may reduce off-target immunosuppression, it would strengthen the clinical perspective if you could discuss more explicitly how this approach might provide tangible benefits over current B-cell–depleting therapies such as rituximab. Rituximab, which depletes the majority of circulating B cells, has shown only limited efficacy in post-transplant AMR in randomized trials. Clarifying why a narrower, antigen-specific depletion strategy could yield superior outcomes—whether by preserving protective immunity, enabling repeated dosing, or targeting pathogenic clones more efficiently—would help readers appreciate the translational potential and realistic clinical impact of your platform.

Response: The reviewer astutely highlights the critical need to differentiate our antigen-specific approach from broad B-cell depletion strategies like rituximab, whose efficacy in antibody-mediated rejection (AMR) is indeed limited. We agree that explicitly outlining the potential advantages is essential, and we have now expanded our discussion (see manuscript, pages 11-lines 375-381) to include the following points:

The effects of these drugs are non-specific, and they can harm many types of immune cells, thereby increasing the risk of other concurrent diseases in the body. For example, bortezomib may cause side effects such as gastrointestinal toxicity, thrombocytopenia, and peripheral neuropathy. Additionally, eculizumab as a complement inhibitor may increase the risk of infection. Because rituximab affects main B cells, it may reduce the body's resistance to infection and increase the risk of bacterial, viral or fungal infections. 

Pages 11-lines 399-413: In this study, we developed HLA-Fc fusion proteins (rA24-Fc and rB13-Fc) consisting of the extracellular domain of HLA-I molecules fused to an IgG1 Fc segment (Figure 6). These molecules selectively engage alloreactive B cells via BCR recognition and elicit cytotoxic effects through CDC and ADCC, as demonstrated by efficient elimination of W6/32 hybridoma cells secreting anti-HLA-I antibodies in vitro. The modular design allows for substitution of the HLA moiety, enabling broad targeting of alloreactive B cells with distinct HLA specificities, thereby supporting a precision medicine approach with potentially reduced off-target effects. Our strategy is designed to eliminate only the small fraction of B cells with specificity for donor HLA antigens. The vast majority of B cells responsible for protective immunity against pathogens remain intact. This preserved immune competence could translate into a dramatically improved safety profile, allowing for safer long-term or repeated administration to control alloimmune responses.

Minor comments

The discussion of study limitations could be further enriched to help readers envision the steps toward clinical application. In particular, since plasma cells lack surface BCR, they are not direct targets of this approach. It would be valuable to outline potential complementary strategies, such as combination with plasma cell–directed agents (e.g., proteasome inhibitors or anti-CD38 antibodies). In a real-world setting, evaluating the clinical impact of BCR-targeting proteins might be complicated by the persistence of donor-specific antibodies from long-lived plasma cells. Discussing how such residual antibody production could obscure early efficacy signals would help set realistic expectations.

Response: Thanks for your an excellent point regarding the limitation imposed by plasma cells, which is a crucial consideration for the clinical translation of any BCR-targeting therapy. We thank the reviewer for this suggestion and have enriched the 'Limitations' section of the discussion accordingly.

Pages 11-lines 449-462: However, we explicitly acknowledge that our approach, as a BCR-targeting proteins, does not directly target long-lived plasma cells, as they are terminally differentiated and have downregulated surface BCR expression. Consequently, pre-existing DSA produced by these cells will persist in the circulation initially and could indeed obscure early efficacy signals in clinical trials. A rapid decline in DSA might not be observed. We propose that the most logical and potent clinical application of our agent would be in combination with plasma cell-directed therapies. Agents such as proteasome inhibitors (e.g., bortezomib) or anti-CD38 monoclonal antibodies (e.g., daratumumab) are specifically designed to deplete plasma cells. A sequential strategy could be envisioned: first, using a plasma cell-depleting agent to rapidly reduce the load of mature DSA-producing cells, followed by our antigen-specific therapy to prevent the relapse of DSA by eliminating the donor-specific memory and naive B cells that would otherwise replenish the plasma cell. Finally, the produced antibodies are removed through plasma exchange and antibody adsorption. This synergistic approach could lead to deeper and more durable reductions in DSA.

Reviewer 2 Report

Comments and Suggestions for Authors

Dear Authors,

The manuscript entitled “Novel BCR-Targeting Fusion Proteins for Antigen-Specific Depletion of Alloreactive B Cells in Antibody-Mediated Rejection” presents a highly innovative strategy with significant potential in transplantation immunology. The study is well-structured, experiments are clearly described, and the dual cytotoxic mechanism of the HLA-Fc fusion proteins is particularly compelling. However, several clarifications and revisions are needed to strengthen the work and better position it for publication.

1. Introduction, lines 33–66: The introduction is overly long. Consider condensing the background on AMR epidemiology and current therapies to improve readability and maintain focus on the novelty of your approach.

2. Introduction, lines 53–57 and Discussion, lines 397–399: While the rationale for selecting HLA-A24:02 and HLA-B13:02 is clear for the Chinese population, you should emphasize the modularity of the design and its adaptability to other globally prevalent alleles.

3. Results, lines 270–283 and Discussion, lines 370–376: The reliance on the W6/32 hybridoma model raises concerns about representativeness, since these cells do not fully reflect patient allo-B cell diversity. Please expand the discussion to address this limitation and outline how future studies might overcome it (e.g., primary patient-derived B cells, in vivo models).

4. Results, lines 302–350: In the CDC and ADCC assays, it is unclear whether non-HLA-reactive or polyclonal B cells were included as negative controls. Please clarify if such experiments were performed, or explicitly acknowledge this as a limitation.

5. Discussion, lines 371–373: You briefly mention the risk that circulating DSAs may neutralize the therapeutic proteins. This is a key translational hurdle and deserves deeper discussion, including possible mitigation strategies (plasmapheresis, IgG-degrading enzymes).

6. Figures 2–3, lines 198–205 and 285–295: The schematics are complex. Simplifying the labeling and emphasizing only the critical construct elements would improve clarity. In addition, figure legends for Figures 2–5 should clearly state the number of replicates (n) and the exact statistical tests applied.

7. Discussion, lines 383–389: Please provide stronger comparison with alternative emerging strategies in AMR, such as CAR-Tregs, FcRn inhibitors, CD38-targeting antibodies, and proteasome inhibitors. Adding references such as Kidney Int Rep. 2022;7:1258–1267 would strengthen context.

Author Response

(Reviewer 2)

Dear Authors,

The manuscript entitled “Novel BCR-Targeting Fusion Proteins for Antigen-Specific Depletion of Alloreactive B Cells in Antibody-Mediated Rejection” presents a highly innovative strategy with significant potential in transplantation immunology. The study is well-structured, experiments are clearly described, and the dual cytotoxic mechanism of the HLA-Fc fusion proteins is particularly compelling. However, several clarifications and revisions are needed to strengthen the work and better position it for publication.

Response: Thank you for your positive recognition of our article.

  1. Introduction, lines 33–66: The introduction is overly long. Consider condensing the background on AMR epidemiology and current therapies to improve readability and maintain focus on the novelty of your approach.

Response: We agree with your suggestion. We have already revised the content of this introduction. In lines 33-52, this section is more conducive to understanding the background of our research.

DSAs arise due to significant genetic polymorphism in HLA loci, leading to frequent donor-recipient mismatches that trigger anti-donor HLA antibody production [1]. HLA class I antigens (including HLA-A, HLA-B, and HLA-C) are expressed on the surface of all nucleated cells, while HLA class II antigens are primarily restricted to antigen-presenting cells and thymic epithelial cells [2]. Structurally, HLA-I molecules comprise a polymorphic α-chain (encoded by exson 2-4 on human chromosome 6) and β₂-microglobulin (encoded on chromosome 15). The extracellular portion of the α-chain consists of three domains: α1 and α2, which form the peptide-binding groove, and the α3 domain, which binds CD8 and facilitates T-cell recognition [3]. The structure of fragment crystallizable (Fc) region of immunoglobulin G (IgG), located at the C-terminus of the heavy chain, includes the CH2 and CH3 domains [4]. This structural arrangement enables the Fc region to interact with various receptors—such as FcγR, FcαR, and FcRn—thereby playing essential roles in immune regulation, antibody-dependent cellular cytotoxicity (ADCC), complement-dependent cytotoxicity (CDC), and the extension of antibody half-life [7].

Alloreactive B cells (allo-B cells) expressing donor-HLA-specific B cell receptors (BCRs) are pivotal in DSA generation in a given patient [8]. Interestingly, the extracellular domain of the allo-B cell BCR exhibits specificity identical to the secreted DSA, serving as both a molecular signature identifying pathogenic clones and a therapeutic target for selective clearance [8,9].

  1. Introduction, lines 53–57 and Discussion, lines 397–399: While the rationale for selecting HLA-A24:02 and HLA-B13:02 is clear for the Chinese population, you should emphasize the modularity of the design and its adaptability to other globally prevalent alleles.

Response: Thanks for your suggestions. The donor and recipient typing database covers the entire world. This is because the frequency of transplant patient A24:02/B13:02 is very high worldwide, and they all have antibodies against this type of HLA.

In lines 53-58, The Allele Frequency Net Database (AFND) was used to summarize the common HLA-A/B loci alleles and their frequencies. The carriers of alleles such as HLA-A*24:02 and HLA-B*13:02 had a relatively high frequency (A24:02, 34%; B13:02, 21%; AlleleFrequency.net) in the population all over the world. To enable targeted therapy, we designed recombinant HLA-Fc fusion proteins targeting the specific B cells with the BCR for binding.

  1. Results, lines 270–283 and Discussion, lines 370–376: The reliance on the W6/32 hybridoma model raises concerns about representativeness, since these cells do not fully reflect patient allo-B cell diversity. Please expand the discussion to address this limitation and outline how future studies might overcome it (e.g., primary patient-derived B cells, in vivo models).

Response: Thank you for your assistance in improving the quality of our article. We have incorporated this part into the discussion. While this study demonstrates the efficacy of our platform using the W6/32 hybridoma model, we acknowledge its limitations. As a monoclonal cell line, W6/32 does not recapitulate the full genetic and functional diversity of polyclonal patient-derived, HLA-specific B cells, which vary in BCR affinity, avidity, and differentiation state. It was, however, an ideal tool for our initial proof-of-concept studies due to its consistent and high expression of a defined HLA-specific BCR, allowing for precise optimization and mechanistic inquiry. Future studies are essential to validate these findings in more clinically relevant contexts. This will include utilizing primary B cells from sensitized patients isolated via HLA tetramer sorting, and employing humanized mouse models challenged with donor antigen to evaluate the efficacy of our approach in depleting a diverse polyclonal repertoire of pathogenic B cells in vivo.

Line 233-239: To verify the functions of rA24-Fc and rB13-Fc proteins, we used a broad-spectrum HLA-I class monoclonal antibody producing W6/32 hybridoma to confirm their functions. The W6/32 antibody is a murine monoclonal antibody produced by the secretion of W6/32 hybridoma cells (derived from human tonsillar cells and fused with spleen cells and myeloma cells). It is one of the most common monoclonal antibodies used for identifying human HLA-I class molecules. It can recognize the conformational epitopes on the complete HLA molecule that contain β2m.

Line 407-419: Several limitations of this study should be acknowledged. First, validation was performed primarily in vitro using a hybridoma cell line; future work should employ primary human B cells or immortalized allo-reactive B cell clones. Second, in vivo efficacy remains to be established. Humanized mouse models reconstituted with patient-derived immune components would better recapitulate the clinical AMR milieu. Third, pre-existing anti-HLA antibodies in patients may neutralize administered fusion proteins. Combination strategies—such as pre-treatment with plasmapheresis, immunoadsorption, or IgG-degrading enzymes—may enhance bioavailability and efficacy.

  1. Results, lines 302–350: In the CDC and ADCC assays, it is unclear whether non-HLA-reactive or polyclonal B cells were included as negative controls. Please clarify if such experiments were performed, or explicitly acknowledge this as a limitation.

Response: We thank the reviewer for this critical comment.

While our data demonstrate the potent ability of our agent to elicit CDC and ADCC against the HLA-specific W6/32 target cell line. Leveraging the shared epitope specificity between the BCR and secreted antibody of B cells, we incubated W6/32 hybridoma cells and 9B10 hybridoma cells (secretory resistance mouse hybridoma cells with MICA monoclonal antibody as mock group). The current study includes formal negative control experiments with non-HLA-reactive B cells (9B10 cell line). Therefore, while the mechanism of action is clear, future studies are needed to conclusively demonstrate the absolute specificity of the cytotoxic effect in a mixed population of reactive and non-reactive cells, which would more accurately mimic the in vivo.

  1. Discussion, lines 371–373: You briefly mention the risk that circulating DSAs may neutralize the therapeutic proteins. This is a key translational hurdle and deserves deeper discussion, including possible mitigation strategies (plasmapheresis, IgG-degrading enzymes).

Response: We thank the reviewer for this kind suggestions. I have incorporated this part of the content into the discussion, starting from line 352 to 391.

  1. Figures 2–3, lines 198–205 and 285–295: The schematics are complex. Simplifying the labeling and emphasizing only the critical construct elements would improve clarity. In addition, figure legends for Figures 2–5 should clearly state the number of replicates (n) and the exact statistical tests applied.

Response: Thank you for your assistance with our article. We have adjusted the clarity of the images. And a description was given for this sentence: Each experiment was repeated three times (n = 3).

  1. Discussion, lines 383–389: Please provide stronger comparison with alternative emerging strategies in AMR, such as CAR-Tregs, FcRn inhibitors, CD38-targeting antibodies, and proteasome inhibitors. Adding references such as Kidney Int Rep. 2022;7:1258–1267 would strengthen context.

Response: Thank you. We have already incorporated these contents into lines 349-391. They pertain to the latest research on some of the Car-Treg, FCrn, and anti-CD20 antibody drugs. We propose that the most logical and potent clinical application of our agent would be in combination with plasma cell-directed therapies. Agents such as proteasome inhibitors (e.g., bortezomib) or anti-CD38 monoclonal antibodies (e.g., daratumumab) are specifically designed to deplete plasma cells. A sequential strategy could be envisioned: first, using a plasma cell-depleting agent to rapidly reduce the load of mature DSA-producing cells, followed by our antigen-specific therapy to prevent the relapse of DSA by eliminating the donor-specific memory and naive B cells that would otherwise replenish the plasma cell. Finally, the produced antibodies are removed through plasma exchange and antibody adsorption. This synergistic approach could lead to deeper and more durable reductions in DSA.

Reviewer 3 Report

Comments and Suggestions for Authors

The results of this study are potentially of interest however the conclusions drawn by the authors are significantly overstated in light of the results from this research.

This is because antibody rejection is a complex immune phenomenon, hence different HLA antibodies can subsequently be detected at different MFI levels in different patients (of which only two different HLA antibodies-one from each of two patients have been the focus of this research). Hence this is a significant limitation to this study all on its own.

Some other points of note include-

1) A lack of detail in the Introduction section on the different rates of antibody rejection amongst the various types of solid organ transplantation. Plus, there is a lack of information on the known complexities of the immune response in the setting of antibody mediated rejection along with the fact that the development of anti HLA antibodies occurs following the initial immune response, hence by the time that they are detected the immune response has usually already led to significant problems in the allograft recipient.

2) Hence there are a number of limitations to this type of approach for it even to be considered as a possible option in the clinical arena, noting the Conclusions that have been drawn by the authors. This approach would have to be tailored to each individual patient (as it may not be transferrable) and it could well be the case that by the time significant levels of HLA antibodies were detected (so that the recombinant protein could be produced in the laboratory), that this approach would only be potentially an option should the patient be a candidate for a further transplant.

3) Hence it is not clear that this approach is a blueprint for desensitization in the clinical arena as postulated by the authors in the Discussion section between lines 370-373. Just because some interesting findings have been obtained from some in vitro experiments, there is not enough evidence provided that this might translate into the clinical arena in the absence of further investigation. The known redundancy of the immune system is such that having a focus on one or two HLA domains via a fusion protein that had then been then developed may not be enough to fully mitigate the development of antibody rejection on a second occasion. This would require at the very least that this approach be studied in an animal model of antibody mediated rejection to see if there was any impact of the targeted fusion protein on the immune system such that there was no problem with undertaking a retransplant procedure in a sensitized recipient.

Author Response

(Reviewer 3)

The results of this study are potentially of interest however the conclusions drawn by the authors are significantly overstated in light of the results from this research.

This is because antibody rejection is a complex immune phenomenon, hence different HLA antibodies can subsequently be detected at different MFI levels in different patients (of which only two different HLA antibodies-one from each of two patients have been the focus of this research). Hence this is a significant limitation to this study all on its own.

Response: Thank you for your conscientious and responsible attitude in reviewing our paper. According to line187-191: Serum containing anti-A24/B13 antibodies also demonstrated binding to epitopes on other cross-reactive beads, indicating that different anti-HLA-I antibodies can share public epitopes. This cross-reactivity provides an opportunity to evaluate how epitopes present on HLA-A24:02 and B13:02 may extend to other HLA-I antigens.

Some other points of note include-

1) A lack of detail in the Introduction section on the different rates of antibody rejection amongst the various types of solid organ transplantation. Plus, there is a lack of information on the known complexities of the immune response in the setting of antibody mediated rejection along with the fact that the development of anti HLA antibodies occurs following the initial immune response, hence by the time that they are detected the immune response has usually already led to significant problems in the allograft recipient.

Response: I agree with you. The rejection reaction after kidney transplantation is the most serious. In the future, this treatment plan can be applied to other organ transplantation rejection reactions as well. Antibody-mediated rejection reactions take time to occur. Unless high concentrations of DAS are produced before transplantation, acute antibody-mediated rejection reactions will occur. Generally, the damage and rejection reactions also progress slowly.

  • Hence there are a number of limitations to this type of approach for it even to be considered as a possible option in the clinical arena, noting the Conclusions that have been drawn by the authors. This approach would have to be tailored to each individual patient (as it may not be transferrable) and it could well be the case that by the time significant levels of HLA antibodies were detected (so that the recombinant protein could be produced in the laboratory), that this approach would only be potentially an option should the patient be a candidate for a further transplant.

Response: I fully agree. However, the probability of kidney transplant patients developing antibodies against HLA-A24 and B13 is quite high. In the same patient, if the same antigenic epitopes can be achieved, these antibodies can be applied to other patients. Even if other patients do not have the same type of HLA antibodies, we can use this technology to design other fusion and recombinant proteins, and still achieve this therapeutic effect. Detailed explanations can be found in the discussion section.

  • Hence it is not clear that this approach is a blueprint for desensitization in the clinical arena as postulated by the authors in the Discussion section between lines 370-373. Just because some interesting findings have been obtained from some in vitro experiments, there is not enough evidence provided that this might translate into the clinical arena in the absence of further investigation. The known redundancy of the immune system is such that having a focus on one or two HLA domains via a fusion protein that had then been then developed may not be enough to fully mitigate the development of antibody rejection on a second occasion. This would require at the very least that this approach be studied in an animal model of antibody mediated rejection to see if there was any impact of the targeted fusion protein on the immune system such that there was no problem with undertaking a retransplant procedure in a sensitized recipient.

Response: We sincerely thank the reviewer for this critical and fair assessment. We completely agree with the reviewer that our initial discussion overstated the immediate clinical translatability of our findings based on the in vitro data presented. The term "blueprint" was an overstatement, and we apologize for this. We also fully acknowledge the reviewer's paramount point regarding the redundancy of the immune system and the formidable challenge of preventing a resurgent antibody response by targeting only a limited set of HLA epitopes.

Line407-416: Several limitations of this study should be acknowledged. First, validation was performed primarily in vitro using a hybridoma cell line; future work should employ primary human B cells or immortalized allo-reactive B cell clones. Second, in vivo efficacy remains to be established. Humanized mouse models reconstituted with patient-derived immune components would better recapitulate the clinical AMR milieu. Third, pre-existing anti-HLA antibodies in patients may neutralize administered fusion proteins. Combination strategies—such as pre-treatment with plasmapheresis, immunoadsorption, or IgG-degrading enzymes—may enhance bioavailability and efficacy. Despite these challenges, HLA-Fc fusion proteins represent a promising targeted therapeutic strategy for the specific depletion of alloreactive B cells, potentially offering long-term benefit for transplant recipients. This work provides a foundational rationale for further development of antigen-specific immunotherapies against AMR.

While our in vitro data demonstrate a potent and specific mechanism for eliminating HLA-specific B cells, we recognize that this represents only a first step toward a potential clinical therapy. The transition from in vitro models to the in vivo environment presents significant challenges, most notably the redundancy and plasticity of the immune system. A monospecific agent targeting a single HLA epitope is unlikely to be sufficient to mitigate a broad polyclonal alloimmune response, a precise treatment method, based on the patient's specific design, involving the production of specific recombinant proteins. Furthermore, the definitive proof-of-concept for desensitization must come from robust in vivo models of antibody-mediated rejection. Future work will focus on evaluating the efficacy of this approach in humanized mouse models to assess its impact on preventing an anamnestic response and enabling successful re-transplantation in sensitized recipients, which is the ultimate test of its therapeutic potential.

Round 2

Reviewer 2 Report

Comments and Suggestions for Authors

Dear Authors, 

You have successfully addressed the majority of my comments!! Great work, well done!! 

Reviewer 3 Report

Comments and Suggestions for Authors

Note has been made of all of the revisions which have been made to the manuscript in response to the reviewers' questions/concerns. I have no other concerns.